# A Limitation of the PAC-Bayes Framework

**Roi Livni**
Department of Electrical Engineering
Tel-Aviv University
Israel
rlivni@tauex.tau.ac.il

**Shay Moran**
Department of Mathematics
Technion, Haifa
Israel
shaymoran1@gmail.com

## Abstract

PAC-Bayes is a useful framework for deriving generalization bounds which was introduced by McAllester ('98). This framework has the flexibility of deriving distribution- and algorithm-dependent bounds, which are often tighter than VC-related uniform convergence bounds. In this manuscript we present a limitation for the PAC-Bayes framework. We demonstrate an easy learning task which is not amenable to a PAC-Bayes analysis.

Specifically, we consider the task of linear classification in 1D; it is well-known that this task is learnable using just $O(\log(1/\delta)/\epsilon)$ examples. On the other hand, we show that this fact can not be proved using a PAC-Bayes analysis: for any algorithm that learns 1-dimensional linear classifiers there exists a (realizable) distribution for which the PAC-Bayes bound is arbitrarily large.

## 1 Introduction

The classical setting of supervised binary classification considers *learning algorithms* that receive (binary) labelled examples and are required to output a *predictor* or a *classifier* that predicts the label of new and unseen examples. Within this setting, Probably Approximately Correct (PAC) generalization bounds quantify the success of an algorithm to approximately predict with high probability. The PAC-Bayes framework, introduced in [22, 34] and further developed in [21, 20, 30], provides PAC-flavored bounds to Bayesian algorithms that produce *Gibbs-classifiers* (also called *stochastic-classifiers*). These are classifiers that, instead of outputting a single classifier, output a probability distribution over the family of classifiers. Their performance is measured by the expected success of prediction where expectation is taken with respect to both sampled data and sampled classifier.

A PAC-Bayes generalization bound relates the generalization error of the algorithm to a KL distance between the stochastic output classifier and some *prior distribution P*. In more detail, the generalization bound is comprised of two terms: first, the empirical error of the output Gibbs-classifier, and second, the KL distance between the output Gibbs classifier and some arbitrary (but sample-independent) prior distribution. This standard bound captures a basic intuition that a good learner needs to balance between bias, manifested in the form of a prior, and fitting the data, which is measured by the empirical loss. A natural task is then, to try and characterize the potential as well as limitations of such Gibbs-learners that are amenable to PAC-Bayes analysis. As far as the potential, several past results established the strength and utility of this framework (e.g. [33, 31, 18, 13, 17]).

In this work we focus on the complementary task, and present the first limitation result showing that there are classes that are learnable, even in the strong distribution-independent setting of PAC, but do not admit any algorithm that is amenable to a non-vacuous PAC-Bayes analysis. We stress that this is true even if we exploit the bound to its fullest and allow any algorithm and any possible, potentially distribution-dependent, prior.

More concretely, we consider the class of 1-dimensional thresholds, i.e. the class of linear classifiers over the real line. It is a well known fact that this class is learnable and enjoys highly optimistic sample complexity. Perhaps surprisingly, though, we show that any Gibbs-classifier that learns the class of thresholds, must output posteriors from an unbounded set. We emphasize that the result is provided even for priors that depend on the data distribution.

From a technical perspective our proof exploits and expands a technique that was recently introduced by Alon et al. [1] to establish limitations on differentially-private PAC learning algorithms. The argument here follow similar lines, and we believe that these similarities in fact highlight a potentially powerful method to derive further limitation results, especially in the context of stability.

## 2 Preliminaries

### 2.1 Problem Setup

We consider the standard setting of binary classification. Let $\mathcal{X}$ denote the domain and $\mathcal{Y} = \{\pm 1\}$ the label space. We study learning algorithms that observe as input a sample $S$ of labelled examples drawn independently from an unknown target distribution $D$, supported on $\mathcal{X} \times \mathcal{Y}$. The output of the algorithm is an hypothesis $h : \mathcal{X} \to \mathcal{Y}$, and its goal is to minimize the $0/1$-loss, which is defined by:

$$\mathcal{L}_D(h) = \mathop{\mathbb{E}}_{(x,y) \sim D} \big[ \mathbf{1}[h(x) \neq y] \big].$$

We will focus on the setting where the distribution $D$ is *realizable* with respect to a fixed hypothesis class $\mathcal{H} \subseteq \mathcal{Y}^{\mathcal{X}}$ which is known in advance. That is, it is assumed that there exists $h \in \mathcal{H}$ such that: $\mathcal{L}_D(h) = 0$. Let $S = \langle (x_1, y_1), \dots, (x_m, y_m) \rangle \in (\mathcal{X} \times \mathcal{Y})^m$ be a sample of labelled examples. The empirical error $\mathcal{L}_S$ with respect to $S$ is defined by

$$\mathcal{L}_S(h) = \frac{1}{m} \sum_{i=1}^{m} \mathbf{1}[h(x) \neq y].$$

We will use the following notation: for a sample $S = \langle (x_1, y_1), \dots (x_m, y_m) \rangle$, let $\underline{S}$ denote the underlying set of unlabeled examples $\underline{S} = \{x_i : i \leq m\}$.

**The Class of Thresholds.**    For $k \in \mathbb{N}$ let $h_k : \mathbb{N} \to \{\pm 1\}$ denote the *threshold function*

$$h_k(x) = \begin{cases} -1 & x \leq k \\ +1 & x > k. \end{cases}$$

The class of thresholds $\mathcal{H}_{\mathbb{N}}$ is the class $\mathcal{H}_{\mathbb{N}} := \{h_k : k \in \mathbb{N}\}$ over the domain $\mathcal{X}_{\mathbb{N}} := \mathbb{N}$. Similarly, for a finite $n \in \mathbb{N}$ let $\mathcal{H}_n$ denote the class of all thresholds restricted to the domain $\mathcal{X}_n := [n] = \{1, \dots, n\}$. Note that $S$ is realizable with respect to $\mathcal{H}_{\mathbb{N}}$ if and only if either (i) $y_i = +1$ for all $i \leq m$, or (ii) there exists $1 \leq j \leq m$ such that $y_i = -1$ if and only if $x_i \leq x_j$.

A basic fact in statistical learning is that $\mathcal{H}_{\mathbb{N}}$ is PAC-learnable. That is, there exists an algorithm $A$ such that for every realizable distribution $D$, if $A$ is given a sample of size $O(\frac{\log 1/\delta}{\epsilon})$ examples drawn from $D$, then with probability at least $1 - \delta$, the output hypothesis $h_S$ satisfies $\mathcal{L}_D(h_S) \leq \epsilon$. In fact, any algorithm $A$ which returns an hypothesis $h_k \in \mathcal{H}_{\mathbb{N}}$ which is consistent with the input sample, will satisfy the above guarantee. Such algorithms are called empirical risk minimizers (ERMs). We stress that the above sample complexity bound is *independent* of the domain size. In particular it applies to $\mathcal{H}_n$ for every $n$, as well as to the infinite class $\mathcal{H}_{\mathbb{N}}$. For further reading, we refer to text books on the subject, such as [32, 23].

### 2.2 PAC-Bayes Bounds

PAC Bayes bounds are concerned with *stochastic-classifiers*, or *Gibbs-classifiers*. A Gibbs-classifier is defined by a distribution $Q$ over hypotheses. The distribution $Q$ is sometimes referred to as a *posterior*. The loss of a Gibbs-classifier with respect to a distribution $D$ is given by the expected loss over the drawn hypothesis and test point, namely:

$$\mathcal{L}_D(Q) = \mathop{\mathbb{E}}_{h \sim Q, (x,y) \sim D} \big[ \mathbf{1}\big[ h(x) \neq y \big] \big].$$

A key advantage of the PAC-Bayes framework is its flexibility of deriving generalization bounds that do not depend on an hypothesis class. Instead, they provide bounds that depend on the KL distance between the output posterior and a fixed prior $P$. Recall that the KL divergence between a distribution $P$ and a distribution $Q$ is defined as follows[1]:

$$\mathrm{KL}\left(P\|Q\right) = \underset{x \sim P}{\mathbb{E}}\Big[\log \frac{P(x)}{Q(x)}\Big].$$

Then, the classical PAC-Bayes bound asserts the following:

**Theorem 1** (PAC-Bayes Generalization Bound [22]). *Let $D$ be a distribution over examples, let $P$ be a prior distribution over hypothesis, and let $\delta > 0$. Denote by $S$ a sample of size $m$ drawn independently from $D$. Then, the following event occurs with probability at least $1 - \delta$: for every posterior distribution $Q$,*

$$\mathcal{L}_D(Q) \leq \mathcal{L}_S(Q) + O\left(\sqrt{\frac{\mathrm{KL}\left(Q\|P\right) + \ln \sqrt{m}/\delta}{m}}\right).$$

The above bound relates the generalization error to the KL divergence between the posterior and the prior. Remarkably, the prior distribution $P$ can be chosen as a function of the target distribution $D$, allowing to obtain distribution-dependent generalization bounds.

Since this pioneer work of McAllester [21], many variations on the PAC-Bayes bounds have been proposed. Notably, Seeger et al. [31] and Catoni [9] provided bounds that are known to converge at rate $1/m$ in the realizable case (see also [15] for an up-to-date survey). We note that our constructions are all provided in the realizable setting, hence readily apply.

## 3 Main Result

We next present the main result in this manuscript. Proofs are provided in the full version [19]. The statements use the following function $\Phi(m, \gamma, n)$, which is defined for $m, n > 1$ and $\gamma \in (0, 1)$:

$$\Phi(m, \gamma, n) = \frac{\log^{(m)}(n)}{(\frac{10m}{\gamma})^{3m}}.$$

Here, $\log^{(k)}(x)$ denotes the iterated logarithm, i.e.

$$\log^{(k)}(x) = \underbrace{\log(\log \ldots (\log(x)))}_{k \text{ times}}.$$

An important observation is that $\lim_{n \to \infty} \Phi(m, \gamma, n) = \infty$ for every fixed $m$ and $\gamma$.

**Theorem 2** (Main Result). *Let $n, m > 1$ be integers, and let $\gamma \in (0, 1)$. Consider the class $\mathcal{H}_n$ of thresholds over the domain $\mathcal{X}_n = [n]$. Then, for any learning algorithm $A$ which is defined on samples of size $m$, there exists a realizable distribution $D = D_A$ such that for any prior $P$ the following event occurs with probability at least $1/16$ over the input sample $S \sim D^m$,*

$$\mathrm{KL}\left(Q_S\|P\right) = \tilde{\Omega}\left(\frac{\gamma^2}{m^2}\log\Big(\frac{\Phi(m, \gamma, n)}{m}\Big)\right) \quad \text{or} \quad \mathcal{L}_D(Q_S) > 1/2 - \gamma - \frac{m}{\Phi(m, \gamma, n)},$$

*where $Q_S$ denotes the posterior outputted by $A$.*

To demonstrate how this result implies a limitation of the PAC-Bayes framework, pick $\gamma = 1/4$ and consider any algorithm $A$ which learns thresholds over the natural numbers $\mathcal{X}_{\mathbb{N}} = \mathbb{N}$ with confidence $1 - \delta \geq 99/100$, error $\epsilon < 1/2 - \gamma = 1/4$, and $m$ examples[2]. Since $\Phi(m, 1/4, n)$ tends to infinity with $n$ for any fixed $m$, the above result implies the existence of a realizable distribution $D_n$ supported on $X_n \subseteq \mathbb{N}$ such that the PAC-Bayes bound with respect to any possible prior $P$ will produce vacuous bounds. We summarize it in the following corollary.

**Corollary 1** (PAC-learnability of Linear classifiers cannot be explained by PAC-Bayes). *Let $\mathcal{H}_{\mathbb{N}}$ denote the class of thresholds over $\mathcal{X}_{\mathbb{N}} = \mathbb{N}$ and let $m > 0$. Then, for every algorithm $A$ that maps inputs sample $S$ of size $m$ to output posteriors $Q_S$ and for every arbitrarily large $N > 0$ there exists a realizable distribution $D$ such that, for any prior $P$, with probability at least $1/16$ over $S \sim D^m$ on of the following holds:*

$$\mathrm{KL}\,(Q_S \| P) > N \qquad \text{or,} \qquad \mathcal{L}_D(Q_S) > 1/4.$$

A different interpretation of Theorem 2 is that in order to derive meaningful PAC-Bayes generalization bounds for PAC-learning thresholds over a finite domain $X_n$, the sample complexity must grow to infinity with the domain size $n$ (it is at least $\Omega(\log^\star(n))$). In contrast, the true sample complexity of this problem is $O(\log(1/\delta)/\epsilon)$ which is independent of $n$.

## 4   Technical Overview

A common approach of proving impossibility results in computer science (and in machine learning in particular) exploits a Minmax principle, whereby one specifies a fixed hard distribution over inputs, and establishes the desired impossibility result for any algorithm with respect to random inputs from that distribution. As an example, consider the "No-Free-Lunch Theorem" which establishes that the VC dimension lower bounds the sample complexity of PAC-learning a class $\mathcal{H}$. Here, one fixes the distribution to be uniform over a shattered set of size $d = \mathsf{VC}(H)$, and argues that every learning algorithm must observe $\Omega(d)$ examples. (See e.g. Theorem 5.1 in [32].)

Such "Minmax" proofs establish a stronger assertion: they apply even to algorithms that "know" the input-distribution. For example, the No-Free-Lunch Theorem applies even to learning algorithms that are designed given the knowledge that the marginal distribution is uniform over some shattered set.

Interestingly, such an approach is bound to fail in proving Theorem 2. The reason is that if the marginal distribution $D_{\mathcal{X}}$ over $\mathcal{X}_n$ is fixed, then one can pick an $\epsilon/2$-cover[3] $\mathcal{C}_n \subseteq \mathcal{H}_n$ of size $|\mathcal{C}_n| = O(1/\epsilon)$, and use any Empirical Risk Minimizer for $\mathcal{C}_n$. Then, by picking the prior distribution $P$ to be uniform over $\mathcal{C}_n$, one obtains a PAC-Bayes bound which scales with the entropy $H(P) = \log|\mathcal{C}_n| = O(\log(1/\epsilon))$, and yields a $\mathsf{poly}(1/\epsilon, \log(1/\delta))$ generalization bound, which is independent of $n$. In other words, in the context of Theorem 2, there is no single distribution which is "hard" for all algorithms.

Thus, to overcome this difficulty one must come up with a "method" which assigns to any given algorithm $A$ a "hard" distribution $D = D_A$, which witnesses Theorem 2 with respect to $A$. The challenge is that $A$ is an arbitrary algorithm; e.g. it may be improper[4] or add different sorts of noise to its output classifier. We refer the reader to [26, 25, 3] for a line of work which explores in detail a similar "failure" of the Minmax principle in the context of PAC learning with low mutual information.

The method we use in the proof of Theorem 2 exploits Ramsey Theory. In a nutshell, Ramsey Theory provides powerful tools which allow to detect, for any learning algorithm, a large *homogeneous* set such that the behavior of $A$ on inputs from the homogeneous set is highly regular. Then, we consider the uniform distribution over the homogeneous set to establish Theorem 2.

We note that similar applications of Ramsey Theory in proving lower bounds in computer science date back to the 80's [24]. For more recent usages see e.g. [8, 11, 10, 1]. Our proof closely follows the argument of Alon et al. [1], which establishes an impossibility result for learning $\mathcal{H}_n$ by differentially-private algorithms.

**Technical Comparison with the Work by Alon et al. [1].**   For readers who are familiar with the work of [1], let us summarize the main differences between the two proofs. The main challenge in extending the technique from [1] to prove Theorem 2 is that PAC-Bayes bounds are only required to hold for *typical samples*. This is unlike the notion of differential-privacy (which was the focus of [1]) that is defined with respect to *all samples*. Thus, establishing a lower bound in the context of differential privacy is easier: one only needs to demonstrate a single sample for which privacy is

breached. However, to prove Theorem 2 one has to demonstrate that the lower bound applies to many samples. Concretely, this affects the following parts of the proof:

(i) The Ramsey argument in the current manuscript (Lemma 1) is more complex: to overcome the above difficulty we needed to modify the coloring and the overall construction is more convoluted.

(ii) Once Ramsey Theorem is applied and the homogeneous subset $R_n \subseteq X_n$ is derived, one still needs to derive a lower bound on the PAC-Bayes quantity. This requires a technical argument (Lemma 2), which is tailored to the definition of PAC-Bayes. Again, this lemma is more complicated than the corresponding lemma in [1].

(iii) Even with Lemma 1 and Lemma 2 in hand, the remaining derivation of Theorem 2 still requires a careful analysis which involves defining several "bad" events and bounding their probabilities. Again, this is all a consequence of that the PAC-Bayes quantity is an "average-case" complexity measure.

## 4.1 Proof Sketch and Key Definitions

The proof of Theorem 2 consists of two steps: (i) detecting a hard distribution $D = D_A$ which witnesses Theorem 2 with respect to the assumed algorithm $A$, and (ii) establishing the conclusion of Theorem 2 given the hard distribution $D$. The first part is combinatorial (exploits Ramsey Theory), and the second part is more information-theoretic. For the purpose of exposition, we focus in this technical overview, on a specific algorithm $\mathcal{A}$. This will make the introduction of the key definitions and presentation of the main technical tools more accessible.

**The algorithm $\mathcal{A}$.** Let $S = \langle (x_1, y_1), \ldots, (x_m, y_m) \rangle$ be an input sample. The algorithm $\mathcal{A}$ outputs the posterior distribution $Q_S$ which is defined as follows: let $h_{x_i} = \mathbf{1}[x > x_i] - \mathbf{1}[x \leq x_i]$ denote the threshold corresponding to the $i$'th input example. The posterior $Q_S$ is supported on $\{h_{x_i}\}_{i=1}^m$, and to each $h_{x_i}$ it assigns a probability according to a decreasing function of its empirical risk. (So, hypotheses with lower risk are more probable.) The specific choice of the decreasing function does not matter, but for concreteness let us pick the function $\exp(-x)$. Thus,

$$Q_S(h_{x_i}) \propto \exp\big(-\mathcal{L}_S(h_{x_i})\big). \tag{1}$$

While one can directly prove that the above algorithm does not admit a PAC-Bayes analysis, we provide here an argument which follows the lines of the general case. We start by explaining the key property of *Homogeneity*, which allows to detect the hard distribution.

### 4.1.1 Detecting a Hard Distribution: Homogeneity

The first step in the proof of Theorem 2 takes the given algorithm and identifies a large subset of the domain on which its behavior is *Homogeneous*. In particular, we will soon see that the algorithm $\mathcal{A}$ is *Homogeneous* on the entire domain $\mathcal{X}_n$. In order to define Homogeneity, we use the following equivalence relation between samples:

**Definition 1** (Equivalent Samples). *Let* $S = \langle (x_1, y_1), \ldots, (x_m, y_m) \rangle$ *and* $S' = \langle (x'_1, y'_1), \ldots, (x'_m, y'_m) \rangle$ *be two samples. We say that $S$ and $S'$ are equivalent if for all $i, j \leq m$ the following holds.*

1. *$x_i \leq x_j \iff x'_i \leq x'_j$, and*

2. *$y_i = y'_i$.*

For example, $\langle (1, -), (5, +), (8, +) \rangle$ and $\langle (10, -), (70, +), (100, +) \rangle$ are equivalent, but $\langle (3, -), (6, +), (4, +) \rangle$ is not equivalent to them (because of Item 1). For a point $x \in \mathcal{X}_n$ let $\mathsf{pos}(x; S)$ denote the number of examples in $S$ that are less than or equal to $x$:

$$\mathsf{pos}(x; S) = \Big| \{ x_i \in \underline{S} : x_i \leq x \} \Big|. \tag{2}$$

For a sample $S = \langle (x_1, y_1), \ldots, (x_m, y_m) \rangle$ let $\pi(S)$ denote the *order-type* of $S$:

$$\pi(S) = (\mathsf{pos}(x_1; S), \mathsf{pos}(x_2; S), \ldots, \mathsf{pos}(x_m; S)). \tag{3}$$

So, the samples $\langle(1,-),(5,+),(8,+)\rangle$ and $\langle(10,-),(70,+),(100,+)\rangle$ have order-type $\pi = (1,2,3)$, whereas $\langle(3,-),(6,+),(4,+)\rangle$ has order-type $\pi = (1,3,2)$.

Note that $S, S'$ are equivalent if and only if they have the same labels-vectors and the same order-type. Thus, we encode the equivalence class of a sample by the pair $(\pi, \bar{y})$, where $\pi$ denotes its order-type and $\bar{y} = (y_1 \ldots y_m)$ denotes its labels-vector. The pair $(\pi, y)$ is called the *equivalence-type of $S$*.

We claim that $\mathcal{A}$ satisfies the following property of *Homogeneity*:

**Property 1** (Homogeneity). *The algorithm $\mathcal{A}$ possesses the following property: for every two equivalent samples $S, S'$ and every $x, x' \in \mathcal{X}_n$ such that $\mathsf{pos}(x, S) = \mathsf{pos}(x', S')$,*

$$\Pr_{h \sim Q_S}[h(x) = 1] = \Pr_{h' \sim Q_{S'}}[h'(x') = 1],$$

*where $Q_S, Q_{S'}$ denote the Gibbs-classifier outputted by $\mathcal{A}$ on the samples $S, S'$.*

In short, Homogeneity means that the probability $h \sim Q_S$ satisfies $h(x) = 1$ depends only on $\mathsf{pos}(x, S)$ and on the equivalence-type of $S$. To see that $\mathcal{A}$ is indeed homogeneous, let $S, S'$ be equivalent samples and let $Q_S, Q_{S'}$ denote the corresponding Gibbs-classifiers outputted by $\mathcal{A}$. Then, for every $x, x'$ such that $\mathsf{pos}(x, S) = \mathsf{pos}(x', S')$, Equation (1) yields that:

$$\Pr_{h \sim Q_S}\left[h(x) = +1\right] = \sum_{x_i < x} Q_S(h_{x_i}) = \sum_{x_i' < x'} Q_{S'}(h_{x_i'}) = \Pr_{h' \sim Q_{S'}}\left[h'(x') = +1\right],$$

where in the second transition we used that $Q_S(h_{x_i}) = Q_{S'}(h_{x_i'})$ for every $i \leq m$ (because $S, S'$ are equivalent), and that $x_i \leq x \iff x_i' \leq x'$, for every $i$ (because $\mathsf{pos}(x, S) = \mathsf{pos}(x', S')$).

**The General Case: Approximate Homogeneity.** Before we continue to define the hard distribution for algorithm $A$, let us discuss how the proof of Theorem 2 handles arbitrary algorithms that are not necessarily homogeneous.

The general case complicates the argument in two ways. First, the notion of Homogeneity is relaxed to an approximate variant which is defined next. Here, an order type $\pi$ is called a *permutation* if $\pi(i) \neq \pi(j)$ for every distinct $i, j \leq m$. (Indeed, in this case $\pi = (\pi(x_1) \ldots \pi(x_m))$ is a permutation of $1 \ldots m$.) Note that the order type of $S = \langle(x_1, y_1) \ldots (x_m, y_m))\rangle$ is a permutation if and only if all the points in $S$ are distinct (i.e. $x_i \neq x_j$ for all $i \neq j$).

**Definition 2** (Approximate Homogeneity). *An algorithm $\mathcal{B}$ is $\gamma$-approximately $m$-homogeneous if the following holds: let $S, S'$ be two equivalent samples of length $m$ whose order-type is a permutation, and let $x \notin \underline{S}, x' \notin \underline{S'}$ such that $\mathsf{pos}(x, S) = \mathsf{pos}(x', S')$. Then,*

$$|Q_S(x) - Q_{S'}(x')| \leq \frac{\gamma}{5m}, \tag{4}$$

*where $Q_S, Q_{S'}$ denote the Gibbs-classifier outputted by $\mathcal{B}$ on the samples $S, S'$.*

Second, we need to identify a sufficiently large subdomain on which the assumed algorithm is approximately homogeneous. This is achieved by the next lemma, which is based on a Ramsey argument.

**Lemma 1** (Large Approximately Homogeneous Sets ). *Let $m, n > 1$ and let $\mathcal{B}$ be an algorithm that is defined over input samples of size $m$ over $\mathcal{X}_n$. Then, there is $\mathcal{X}' \subseteq \mathcal{X}_n$ of size $|\mathcal{X}'| \geq \Phi(m, \gamma, n)$ such that the restriction of $\mathcal{B}$ to input samples from $\mathcal{X}'$ is $\gamma$-approximate $m$-homogeneous.*

We prove Lemma 1 in the full version [19]. For the rest of this exposition we rely on Property 1 as it simplifies the presentation of the main ideas.

**The Hard Distribution $D$.** We are now ready to finish the first step and define the "hard" distribution $D$. Define $D$ to be uniform over examples $(x, y)$ such that $y = h_{n/2}(x)$. So, each drawn example $(x, y)$ satisfies that $x$ is uniform in $\mathcal{X}_n$ and $y = -1$ if and only if $x \leq n/2$. In the general case, $D$ will be defined in the same way with respect to the detected homogeneous subdomain.

### 4.1.2 Hard Distribution $\implies$ Lower Bound: Sensitivity

We next outline the second step of the proof, which establishes Theorem 2 using the hard distribution $D$. Specifically, we show that for a sample $S \sim D^m$,

$$\mathrm{KL}\left(Q_S \| P\right) = \tilde{\Omega}\left(\frac{1}{m^2} \log(|\mathcal{X}_n|)\right),$$

with a constant probability bounded away from zero. (In the general case $|X_n|$ is replaced by $\Phi(m, \gamma, n)$ – the size of the homogeneous set.)

**Sensitive Indices.** We begin with describing the key property of homogeneous learners. Let $(\pi, \bar{y})$ denote the equivalence-type of the input sample $S$. By homogeneity (Property 1), there is a list of numbers $p_0, \ldots, p_m$, which depends only on the order-type $(\pi, \bar{y})$, such that $\mathrm{Pr}_{h \sim Q_S}[h(x) = 1] = p_i$ for every $x \in \mathcal{X}_n$, where $i = \mathsf{pos}(x, S)$. The crucial observation is that there exists an index $i \le m'$ which is *sensitive* in the sense that

$$p_i - p_{i-1} \ge \frac{1}{m}. \tag{5}$$

Indeed, consider $x_j$ such that $h_{x_j} = \arg\min_k \mathcal{L}_S(h_{x_k})$, and let $i = \mathsf{pos}(x_j, S)$. Then,

$$p_i - p_{i-1} = \frac{\mathcal{L}_S(h_{x_j})}{\sum_{i' \le m} \mathcal{L}_S(h_{x_{i'}})} \ge \frac{1}{m}.$$

In the general case we show that any homogeneous algorithm that learns $\mathcal{H}_n$ satisfies Equation (5) for *typical* samples (see the full version [19]). The intuition is that any algorithm that learns the distribution $D$ must output a Gibbs-classifier $Q_S$ such that for typical points $x$, if $x > n/2$ then $\mathrm{Pr}_{h \sim Q_S}[h(x) = 1] \approx 1$, and if $x \le n/2$ then $\mathrm{Pr}_{h \sim Q_S}[h(x) = 1] \approx 0$. Thus, when traversing all $x$'s from 1 up to $n$ there must be a jump between $p_{i-1}$ and $p_i$ for some $i$.

**From Sensitive Indices to a Lower Bound on the KL-divergence.** How do sensitive indices imply a lower bound on PAC-Bayes? This is the most technical part of the proof. The crux of it is a connection between sensitivity and the KL-divergence which we discuss next. Consider a sensitive index $i$ and let $x_j$ be the input example such that $\mathsf{pos}(x_j, S) = i$. For $\hat{x} \in \mathcal{X}_n$, let $S_{\hat{x}}$ denote the sample obtained by replacing $x_j$ with $\hat{x}$:

$$S_{\hat{x}} = \langle (x_1, y_1), \ldots, (x_{j-1}, y_{j-1}), (\hat{x}_j, y_j), (x_{j+1}, y_{j+1}) \ldots (x_m, y_m).\rangle,$$

and let $Q_{\hat{x}} := Q_{S_{\hat{x}}}$ denote the posterior outputted by $\mathcal{A}$ given the sample $S_{\hat{x}}$. Consider the set $I \subseteq \mathcal{X}_n$ of all points $\hat{x}$ such that $S_{\hat{x}}$ is equivalent to $S$. Equation (5) implies that that for every $x, \hat{x} \in I$,

$$\Pr_{h \sim Q_{\hat{x}}}[h(x) = 1] = \begin{cases} p_{i-1} & x < \hat{x}, \\ p_i & x > \hat{x}. \end{cases}$$

Combined with the fact that $p_i - p_{i-1} \ge 1/m$, this implies a lower bound on KL-divergence between an arbitrary prior $P$ and $Q_{\hat{x}}$ for most $\hat{x} \in I$. This is summarized in the following lemma:

**Lemma 2** (Sensitivity Lemma). *Let $I$ be a linearly ordered set and let $\{Q_{\hat{x}}\}_{\hat{x} \in I}$ be a family of posteriors supported on $\{\pm 1\}^I$. Suppose there are $q_1 < q_2 \in [0, 1]$ such that for every $x, \hat{x} \in I$:*

$$x < \hat{x} \implies \Pr_{h \sim Q_{\hat{x}}}[h(x) = 1] \le q_1 + \frac{q_2 - q_1}{4},$$

$$x > \hat{x} \implies \Pr_{h \sim Q_{\hat{x}}}[h(x) = 1] \ge q_2 - \frac{q_2 - q_1}{4}.$$

*Then, for every prior distribution $P$, if $\hat{x} \in I$ is drawn uniformly at random, then the following event occurs with probability at least $1/4$:*

$$\mathrm{KL}\left(Q_{\hat{x}} \| P\right) = \Omega\left((q_2 - q_1)^2 \frac{\log|I|}{\log\log|I|}\right).$$

The sensitivity lemma tells us that in the above situation, the KL divergence between $Q_{\hat{x}}$ and any prior $P$, for a random choice $\hat{x}$, scales in terms of two quantities: the distance between the two values, $q_2 - q_1$, and the size of $I$.

The proof of Lemma 2 is provided in the full version [19]. In a nutshell, the strategy is to bound from below $\mathrm{KL}\left(Q_{\hat{x}}^r \| P^r\right)$, where $r$ is sufficiently small; the desired lower bound then follows from the chain rule, $\mathrm{KL}\left(Q_{\hat{x}} \| P\right) = \frac{1}{r}\mathrm{KL}\left(Q_{\hat{x}}^r \| P^r\right)$. Obtaining the lower bound with respect to the $r$-fold products is the crux of the proof. In short, we will exhibit events $E_{\hat{x}}$ such that $Q_{\hat{x}}^r(E_{\hat{x}}) \geq \frac{1}{2}$ for every $\hat{x} \in I$, but $P^r(E_{\hat{x}})$ is tiny for $\frac{|I|}{4}$ of the $\hat{x}$'s. This implies a lower bound on $\mathrm{KL}\left(Q_{\hat{x}}^r \| P^r\right)$ since

$$\mathrm{KL}\left(Q_{\hat{x}}^r \| P^r\right) \geq \mathrm{KL}\left(Q_{\hat{x}}^r(E_{\hat{x}}) \| P^r(E_{\hat{x}})\right),$$

by the data-processing inequality.

**Wrapping Up.** We now continue in deriving a lower bound for $\mathcal{A}$. Consider an input sample $S \sim D^m$. In order to apply Lemma 2, fix any equivalence-type $(\pi, y)$ with a sensitive index $i$ and let $x_j$ be such that $\mathrm{pos}(x_j; S) = i$. The key step is to condition the random sample $S$ on $(\pi, y)$ as well as on $\{x_t\}_{t=1}^m \setminus \{x_j\}$ – all sample points besides the sensitive point $x_j$. Thus, only $x_j$ is remained to be drawn in order to fully specify $S$. Note then, that by symmetry $\hat{x}$ is uniformly distributed in a set $I \subseteq \mathcal{X}_n$, and plugging $q_1 := p_i, q_2 := p_{i-1}$ in Lemma 2 yields that for any prior distribution $P$:

$$\mathrm{KL}\left(Q_S \| P\right) \geq \tilde{\Omega}\left(\frac{1}{m^2}\log(|I|)\right),$$

with probability at least $1/4$. Note that we are not quite done since the size $|I|$ is a random variable which depends on the type $(\pi, \bar{y})$ and the sample points $\{x_k\}_{k \neq j}$. However, the distribution of $|I|$ can be analyzed by elementary tools. In particular, we show that $|I| \geq \Omega(|\mathcal{X}_n|/m^2)$ with high enough probability, which yields the desired lower bound on the PAC-Bayes quantity. (In the general case $|\mathcal{X}_n|$ is replaced by the size of the homogeneous set.)

## 5 Discussion

In this work we presented a limitation for the PAC-Bayes framework by showing that PAC-learnability of one-dimensional thresholds can not be established using PAC-Bayes.

Perhaps the biggest caveat of our result is the mild dependence of the bound on the size of the domain in Theorem 2. In fact, Theorem 2 does not exclude the possibility of PAC-learning thresholds over $\mathcal{X}_n$ with sample complexity that scale with $O(\log^* n)$ such that the PAC-Bayes bound vanishes. It would be interesting to explore this possibility; one promising direction is to borrow ideas from the differential privacy literature: [4] and [6] designed a private learning algorithm for thresholds with sample complexity $\exp(\log^* n)$; this bound was later improved by [16] to $\tilde{O}((\log^* n)^2)$. Also, [7] showed that finite Littlestone dimension is sufficient for private learnability, and it would be interesting to extend these results to the context of PAC-Bayes. Let us note that in the context of *pure* differential privacy, the connection between PAC-Bayes analysis and privacy has been established in [14].

**Non-uniform learning bounds** Another aspect is the implication of our work to learning algorithms beyond the uniform PAC setting. Indeed, many successful and practical algorithms exhibit sample complexity that depends on the target-distribution. E.g.,the $k$-Nearest-Neighbor algorithm eventually learns any target-distribution (with a distribution-dependent rate). The first point we address in this context concerns *interpolating algorithms*. These are learners that achieve zero (or close to zero) training error (i.e. they interpolate the training set). Examples of such algorithms include kernel machines, boosting, random forests, as well as deep neural networks [5, 29]. PAC-Bayes analysis has been utilized in this context, for example, to provide margin-dependent generalization guarantees for kernel machines [18]. It is therefore natural to ask whether our lower bound has implications in this context. As a simple case-study, consider the 1-Nearest-Neighbour. Observe that this algorithm forms a proper and consistent learner for the class of 1-dimensional thresholds[5], and therefore enjoys a very fast learning rate. On the other hand, our result implies that for any

algorithm (including as 1-Nearest-Neighbor) that is amenable to PAC-Bayes analysis, there is a distribution realizable by thresholds on which it has high population error. Thus, no algorithm with a PAC-Bayes generalization bound can match the performance of nearest-neighbour with respect to such distributions.

Finally, this work also relates to a recent attempt to explain generalization through the implicit bias of learning algorithms: it is commonly argued that the generalization performance of algorithms can be explained by an implicit algorithmic bias. Building upon the flexibility of providing distribution-dependent generalization bounds, the PAC-Bayes framework has seen a resurgence of interest in this context towards explaining generalization in large-scale modern-time practical algorithms [27, 28, 13, 14, 2]. Indeed PAC-Bayes bounds seem to provide non-vacuous bounds in several relevant domains [17, 14]. Nevertheless, the work here shows that any algorithm that can learn 1D thresholds is necessarily not biased, in the PAC-Bayes sense, towards a (possibly distribution-dependent) prior. We mention that recently, [12] showed that SGD's generalization performance indeed cannot be attributed to some implicit bias of the algorithm that governs the generalization.

## Broader Impact

There are no foreseen ethical or societal consequences for the research presented herein.

## Acknowledgments and Disclosure of Funding

R.L is supported by an ISF grant no. 2188/20 and partially funded by an unrestricted gift from Google. Any opinions, findings, and conclusions or recommendations expressed in this work are those of the author(s) and do not necessarily reflect the views of Google. S.M is supported by the Israel Science Foundation (grant No. 1225/20), by an Azrieli Faculty Fellowship, and by a grant from the United States - Israel Binational Science Foundation (BSF). Part of this work was done while the author was at Google Research.

## Footnotes

[1]We use here the standard convention that if $P(\{x : Q(x) = 0\}) > 0$ then $\mathrm{KL}\left(P\|Q\right) = \infty$.

[2]We note in passing that any Empirical Risk Minimizer learns thresholds with these parameters using $< 50$ examples.

[3]I.e. $\mathcal{C}_n$ satisfies that $(\forall h \in \mathcal{H}_n)(\exists c \in \mathcal{C}_n) : \Pr_{x \sim D_X}(c(x) \neq h(x)) \leq \epsilon/2$.

[4]I.e. $A$ may output hypotheses which are not thresholds, or Gibbs-classifiers supported on hypotheses which are not thresholds.

[5]Indeed, given any realizable sample it will output the threshold which maximizes the margin.

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
