[Supplementary Material]

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

, $\text{KL}(Q_{\hat{x}} \| P) = \frac{1}{r}\text{KL}(Q_{\hat{x}}^r \| P^r)$. Obtaining the lower bound with respect to the $r$-fold products is the crux of the proof. In short, we will exhibit events $E_{\hat{x}}$ such that $Q_{\hat{x}}^r(E_{\hat{x}}) \geq \frac{1}{2}$ for every $\hat{x} \in I$, but $P^r(E_{\hat{x}})$ is tiny for $\frac{|I|}{4}$ of the $\hat{x}$'s. This implies a lower bound on $\text{KL}(Q_{\hat{x}}^r \| P^r)$ since

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

[6]For concreteness, let $i$ be the minimal sensitive index.

[7]We use here the convention, that $\frac{\log x}{\log \log x} = -\infty$ for $x \leq 2$. Alternatively, one can assume that Equation (7) holds vacuously if $|I(S)| = 0$

[8]Here we assume without loss of generality that $p_{i-1}^{(\pi, \bar{y})} < p_i^{(\pi, \bar{y})}$. If the reverse inequality holds then the argument follows by applying Lemma 2 with respect to the reverse linear order over $I(T)$.

[9] A subset of the universe is homogeneous if all of its $t$-subsets have the same color.

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

# A  Proofs

## A.1  Proof of Theorem 2

Let $A$ be an algorithm as in the premise of Theorem 2. That is, $A$ receives as input a labeled sample $S$ of length $m$ and outputs a posterior $Q_S$. By Lemma 1, there exists $\mathcal{X}' \subseteq \mathcal{X}_n$ of size

$|\mathcal{X}'| = k \geq \Phi(m, \gamma, n)$ such that the restriction of $A$ to inputs from $\mathcal{X}'$ is $\gamma$-approximate $m$-homogeneous. Without loss of generality, assume that $\mathcal{X}' = \mathcal{X}_k$ consists of the first $k$ points in $\mathcal{X}_n$ and that $k$ is an even number.

By the definition of approximate homogeneity (Definition 2) it follows that for every equivalence type $(\pi, \bar{y})$, where $\pi$ is a permutation, there is a list $(p_i^{(\pi, \bar{y})})_{i=0}^m \in [0, 1]^{m+1}$ such that for every sample $S \in (\mathcal{X}_k \times \{0, 1\})^m$ whose type is $(\pi, \bar{y})$ and and every $x \in \mathcal{X}_k \setminus S$:

$$\left| Q_S(x) - p_i^{(\pi, \bar{y})} \right| \leq \frac{\frac{\gamma}{5m}}{2} = \frac{\gamma}{10m},$$

where $\mathsf{pos}(x, S) = i$. For the rest of the proof fix $D$ to be the distribution over examples $(x, y)$ such that $x$ is drawn uniformly from $\mathcal{X}_k$ and $y = -1$ if and only if $x \leq k/2$. The underlying property we will require is summarized in the following claim:

**Claim 1.** *Let $(\pi, \bar{y})$ be an equivalence-type, where $\pi$ is a permutation. Then, one of the following holds: either there exists a sensitive index $0 \leq i \leq m$ such that*

$$|p_i^{(\pi, \bar{y})} - p_{i-1}^{(\pi, \bar{y})}| \geq \frac{\gamma}{2m}, \tag{6}$$

*or else,*

$$\mathcal{L}_D(Q_S) > \frac{1}{2} - \gamma - \frac{m}{k}$$

*with probability 1 over $S \sim D^m(\cdot | (\pi, \bar{y}))$.*

The proof of Claim 1 is deterred to Appendix A.1.1.

With Claim 1 in hand we proceed with the proof of Theorem 2. Let $S$ be a sample and let $(\pi, \bar{y})$ denote its equivalence-type. Define an interval $I(S) \subseteq \mathcal{X}_k$ as follows.

- if $\pi$ is not a permutation then $I(S) = \emptyset$.
- If $(\pi, \bar{y})$ does not have a sensitive index that satisfies Equation (6) then $I(S) = \emptyset$.
- Finally, if $\pi$ is a permutation and $(\pi, \bar{y})$ has a sensitive index $i$ then set[6]

$$I(S) = \begin{cases} (x_j^-, x_j^+) & \frac{k}{2} \notin (x_j^-, x_j^+), \\ (x_j^-, \frac{k}{2}] & \frac{k}{2} \in (x_j^-, x_j^+) \text{ and } y_j = -1, \\ (\frac{k}{2}, x_j^+) & \frac{k}{2} \in (x_j^-, x_j^+) \text{ and } y_j = +1, \end{cases}$$

where $x_j$ is such that $\mathsf{pos}(x_j; S) = i$, and $x_j^- = \max(\{x_t : x_t < x_j\} \cup \{0\})$ and $x_j^+ = \min(\{x_t : x_t > x_j\} \cup \{k+1\})$.

We next define two events which will be used to finish the proof. First, consider the event that the drawn sample $S$ satisfies either[7]

$$\mathrm{KL}(Q_S \| P) = \Omega\left( \frac{\gamma^2}{m^2} \frac{\log |I(S)|}{\log \log |I(S)|} \right), \tag{7}$$

or

$$\mathcal{L}(Q_S) \geq \frac{1}{2} - \gamma - \frac{m}{k}, \tag{8}$$

We show that this event occurs with probability at least $1/4$:

**Claim 2.** *Define $E_1$ to be the event*

$$E_1 = \left\{ S \in (\mathcal{X}_k \times \{\pm 1\})^m : S \text{ satisfies Equation (7) or Equation (8)} \right\}.$$

*Then, $E_1$ occurs with probability at least $1/4$ over $S \sim D^m$.*

The proof of Claim 2 is deterred to Appendix A.1.2. The second event we consider is that the drawn sample $S$ satisfies either Equation (8) or

$$|I(S)| \geq \frac{\Phi(m, \gamma, n)}{8(m+1)^2}. \tag{9}$$

We show that this event occurs with probability at least $7/8$:

**Claim 3.** *Define $E_2$ to be the event*

$$E_2 = \{S : S \text{ satisfies Equation (8) or Equation (9)}\}.$$

*Then $E_2$ occurs with probability at least $7/8$ over $S \sim D^m$*

The proof of Claim 3 is deterred to Appendix A.1.3. With Claims 2 and 3 in hand, the proof of Theorem 2 is completed as follows. First, a union bound implies that the event $E_1 \cap E_2$ occurs with probability at least $1/16$. That is, with probability at least $1/16$ either Equation (8) holds and we are done, or else, if Equation (8) doesn't hold, then both Equations (7) and (9) hold simultaneously, which yields that

$$\mathrm{KL}\left(Q_S \| P\right) \geq \Omega\left(\frac{\gamma^2}{m^2} \frac{\log|I(S)|}{\log\log|I(S)|}\right) \qquad \text{(By Equation (7))}$$

$$\geq \Omega\left(\frac{\gamma^2}{m^2} \frac{\log \frac{\Phi(m,\gamma,n)}{8(m+1)^2}}{\log\log \frac{\Phi(m,\gamma,n)}{8(m+1)^2}}\right). \qquad \text{(By Equation (9))}$$

This concludes the proof of Theorem 2.

$\square$

We are thus left with proving Claims 1 to 3.

### A.1.1 Proof of Claim 1

Let $(\pi, \bar{y})$ be an equivalence-type such that $\pi$ is a permutation. Assume that

$$\mathcal{L}(Q_S) < \frac{1}{2} - \gamma - \frac{m}{k} \tag{10}$$

occurs with a positive probability over $S \sim D^m(\cdot|\pi, \bar{y})$. We first show that there is $i$ such that

$$|p_i^{\pi,\bar{y}} - p_0^{\pi,\bar{y}}| > \gamma/2. \tag{11}$$

Indeed, assume the contrary and fix a sample $S$ with type $(\pi, \bar{y})$ which satisfies Equation (10). Recall that $A$ is homogeneous, hence for every $x \notin \underline{S}$,

$$|Q_S(x) - p_i^{\pi,\bar{y}}| < \frac{\gamma}{10m},$$

where $i = \mathsf{pos}(x, S)$. On the other hand, since Equation (11) is not met by any $i$, it follows that for every $x \notin \underline{S}$:

$$\begin{aligned}
|Q_S(x) - p_0^{\pi,\bar{y}}| &= |Q_S(x) - p_i^{\pi,\bar{y}} + p_i^{\pi,\bar{y}} - p_0^{\pi,\bar{y}}| \\
&\leq |Q_S(x) - p_i^{\pi,\bar{y}}| + |p_i^{\pi,\bar{y}} - p_0^{\pi,\bar{y}}| \\
&\leq \frac{\gamma}{10m} + \frac{\gamma}{2} \\
&\leq \gamma.
\end{aligned}$$

Thus, $\Pr_{h \sim Q_S}[h(x) = 1] \in [p_0^{\pi,\bar{y}} - \gamma, p_0^{\pi,\bar{y}} + \gamma]$, for every $x \in \mathcal{X}_k \setminus \underline{S}$. Now, since $\Pr_{(x,y) \sim D}[y = 1] = 1/2$ it follows that

$$\mathcal{L}_D(Q_S) \geq \frac{1}{2} - \gamma - \frac{m}{k}.$$

Indeed, for every $x \notin S$, if $x \leq k/2$ then $h \sim Q_S$ errs on $x$ with probability at least $q_1 = p_0^{\pi,\bar{y}} - \gamma$, and if $x > k/2$ then $h \sim Q_S$ errs on $x$ with probability at least $q_2 = 1 - (p_0^{\pi,\bar{y}} + \gamma)$. Thus, the

expected loss of $h \sim Q_S$ conditioned on $x \notin \underline{S}$ is at least $\frac{q_1 + q_2}{2} = 1/2 - \gamma$, and the above inequality follows by taking into account that $h \sim Q_S$ may have zero error on the $m$ points in $S$.

Finally, let $i$ be some index that satisfy Equation (11), then because $0 \leq i \leq m$ we obtain via telescoping that there must be some $i' \leq i$, such that

$$|p_{i'}^{\pi, \bar{y}} - p_{i'-1}^{\pi, \bar{y}}| \geq \frac{\gamma}{2m}.$$

$\square$

### A.1.2 Proof of Claim 2

***Proof of Claim 2.*** It is enough to show that $E_1$ occurs with probability at least $1/4$ over $S \sim D^m(\cdot|\pi, \bar{y})$ for every fixed equivalence-type $(\pi, \bar{y})$. Indeed, by summing over all equivalence types, the law of total probability then implies that $E_1$ occurs with probability at least $1/4$ over $S \sim D^m$.

Fix an equivalence-type $(\pi, \bar{y})$. We may assume that $\pi$ is a permutation and that $(\pi, \bar{y})$ has a sensitive index $i$ (or else Equation (7) trivially holds by the definition of $I(S)$ and we are done). If Equation (8) holds with probability at least $1/4$ then also $E$ occurs with probability at least $1/4$ and we are done. Thus, assume that Equation (8) holds with probability less than $1/4$. It suffices to show that Equation (7) holds with probability at least $1/4$. By Claim 1, there is a sensitive index $i$ such that

$$|p_i^{(\pi, \bar{y})} - p_{i-1}^{(\pi, \bar{y})}| > \frac{\gamma}{2m}.$$

Let $x_j$ in $S$ be such that $\mathsf{pos}(x_j; S) = i$. It will be convenient to consider the following (slightly convoluted) process of sampling a pair of (correlated) samples from $D^m(\cdot|\pi, \bar{y})$:

1. Sample $T = \langle (x_1, y_1) \ldots (x_m, y_m) \rangle \sim D^m(\cdot|\pi, \bar{y})$.

2. Resample only the sensitive point $x_j$ while keeping all other points fixed, as well as the equivalence type $(\pi, \bar{y})$. Let $\hat{x}$ denote the newly sampled point and let $T_{\hat{x}}$ denote the sample obtained by replacing $x_j$ by $\hat{x}$.

3. Set $S = T_{\hat{x}}$

Note that both $T$ and $S$ are drawn from $D^m(\cdot|\pi, \bar{y})$ and that $I(T) = I(S)$ always. Since the marginal distribution of $D$ is uniform over $\mathcal{X}_k$, by symmetry it follows that the point $\hat{x}$ drawn in Step 2 is uniform in the interval $I(T) = I(S)$. Our next step is to apply Lemma 2 on the family of distributions $\{Q_{T_{\hat{x}}}\}_{\hat{x} \in I(T)}$. Towards this end, we first fix $T$ and show that the premise of Lemma 2 is satisfied, with $I = I(T)$, $q_1 = p_{i-1}^{(\pi, \bar{y})}$ and $q_2 = p_i^{(\pi, \bar{y})}$.[8] Indeed, by homogeneity it follows that for each $x \in I(T)$, if $x < \hat{x}$

$$\left| \Pr_{h \sim Q_{\hat{x}}}[h(x) = 1] - p_{i-1}^{\pi, \bar{y}} \right| \leq \frac{\gamma}{10m}$$

$$< \frac{|p_i^{(\pi, \bar{y})} - p_{i-1}^{(\pi, \bar{y})}|}{4}, \qquad \text{(because } i \text{ is sensitive)}$$

and similarly if $x \geq \hat{x}$:

$$\left| \Pr_{h \sim Q_{\hat{x}}}[h(x) = 1] - p_i^{(\pi, \bar{y})} \right| < \frac{|p_i^{(\pi, \bar{y})} - p_{i-1}^{(\pi, \bar{y})}|}{4}.$$

Thus, applying Lemma 2 on the family of distributions $\{Q_{T_{\hat{x}}}\}_{\hat{x}\in I(T)}$ yields that for every $T$ sampled in Step 1, the following holds with probability at least $1/4$ over sampling $\hat{x}$:

$$\mathrm{KL}\left(Q_S\|P\right) = \mathrm{KL}\left(Q_{T_{\hat{x}}}\|P\right)$$
$$\geq \Omega\left(\left(p_{i-1}^{(\pi,\bar{y})} - p_i^{(\pi,\bar{y})}\right)^2 \frac{\log(|I(T)|)}{\log\log|I(T)|}\right)$$
$$\geq \Omega\left(\frac{\gamma^2}{m^2}\frac{\log|I(T)|}{\log\log|I(T)|}\right)$$
$$= \Omega\left(\frac{\gamma^2}{m^2}\frac{\log|I(S)|}{\log\log|I(S)|}\right).$$

Note that the above holds for any fixed $T$. Taking expectation over $T$ it follows that with probability at least $1/4$ over $S \sim D(\cdot|(\pi,\bar{y}))$,

$$\mathrm{KL}\left(Q_S\|P\right) \geq \Omega\left(\frac{\gamma^2}{m^2}\frac{\log|I(S)|}{\log\log|I(S)|}\right).$$

As discussed, taking expectation over the equivalence type concludes the proof. □

### A.1.3 Proof of Claim 3

***Proof of Claim 3.*** Consider $S \sim D^m$ where $S = \langle(x_1,y_1),\ldots(x_m,y_m)\rangle$. We claim that with probability at least $7/8$, every two unlabeled examples $x_i, x_j$ with $i \neq j$ are at distance at least $\frac{k}{8(m+1)^2}$ from each other and from $k/2$. Indeed, fix any distinct $x', x'' \in \{x_1,\ldots,x_m,k/2\}$. Recall that the distribution $D$ satisfies that $x_1,\ldots,x_m$ are sampled uniformly and ind. from $\mathcal{X}_k$. Thus, the probability that $0 \leq x' - x'' < \frac{k}{8(m+1)^2}$ is at most $\frac{1}{8(m+1)^2}$. A union bound over all possible $\binom{m+1}{2}$ pairs implies that that the following holds with probability at least $\frac{7}{8}$ over $S \sim D^m$:

$$\left(\forall \text{ distinct } x', x'' \in \left\{x_1,\ldots,x_m,\frac{k}{2}\right\}\right) : |x' - x''| \geq \frac{k}{8(m+1)^2}. \tag{12}$$

We will now show that the latter event implies $E_2$. Let $S$ be a sample satisfying Equation (12). In particular, $x_i \neq x_j$ for every distinct $i,j \leq m$ and so the order-type $\pi = \pi(S)$ is a permutation. Now, if $S$ satisfies Equation (8) then $S \in E_2$ and we are done. Else, by Claim 1 there exists a sensitive index that satisfies Equation (6) and therefore $I(S) = (x', x'')$, where $x', x''$ are distinct points in $\{x_1\ldots,x_m,k/2\}$. Thus,

$$|I(S)| \geq \frac{k}{8(m+1)^2},$$

and Equation (9) holds, which also gives $S \in E_2$. Thus, every $S$ which satisfies Equation (12) is in $E_2$ and so $E_2$ occurs with probability at least $7/8$. □

## A.2 Proof of Lemma 1

We next prove Lemma 1 which establishes the existence of a "largish" homogeneous set with respect to an arbitrary algorithm $A$.

**Notation.** Recall from Equation (2) the definition of $\mathsf{pos}(x,S)$ which was defined for a sample $S$ and a point $x$. It will be convenient to extend this definition to sets: for $R \subseteq \mathcal{X}_n$ and $x \in \mathcal{X}_n$ define $\mathsf{pos}(x,R) = |\{x' \in R : x' \leq x\}|$.

**From Sets to Samples.** Let $(\pi, \bar{y})$ be an equivalence-type whose order-type is a permutation and let $D = \{x_1 < \ldots < x_m\} \subseteq \mathcal{X}_n$ be a set of $m$ points. Denote by $D^{\pi,\bar{y}} = \langle(x_{i_j}, y_{i_j})\rangle_{j=1}^m$ the sample obtained by ordering and labeling the elements of $D$ such that $D^{\pi,\bar{y}}$ has type $(\pi,\bar{y})$; that is, $D^{\pi,\bar{y}}$ is defined such that for every $j \leq m$,

$$\pi(j) = \mathsf{pos}(x_{i_j}, D^{\pi,\bar{y}}) = \mathsf{pos}(x_{i_j}, D) \text{ and } \bar{y} = (y_1,\ldots,y_m). \tag{13}$$

**A Coloring.** We define a coloring over subsets $D \subseteq \mathcal{X}_n$ of size $|D| = m + 1$. Let $D = \{x_0 < x_1 < \ldots < x_m\}$ be a $(m+1)$-subset of $\mathcal{X}_n$. The coloring assigned to $D$ is

$$c(D) = \left\{ (p_0^{\pi,\bar{y}}, \ldots, p_m^{\pi,\bar{y}}) : (\pi, \bar{y}) \text{ is an equivalence-type s.t. } \pi \text{ is a permutation} \right\},$$

where each $p_i^{\pi,\bar{y}}$ is defined as follows: let $D_{-i} = D \setminus \{x_i\}$. For each equivalence type $(\pi, \bar{y})$ such that $\pi$ is a permutation consider the sample $D_{-i}^{\pi,\bar{y}}$ (see Equation (13)), and define $p_i^{\pi,\bar{y}}$ to be the fraction of the form $\frac{t \cdot \gamma}{10m}$ for $t \in \mathbb{N}$ which is closest to

$$\Pr_{h \sim Q_{-i}^{\pi,\bar{y}}} [h(x_i) = 1],$$

where $Q_{-i}^{\pi,\bar{y}}$ is the stochastic classifier obtained by applying $A$ on $D_{-i}^{\pi,\bar{y}}$.

Since the total number of equivalence-types whose order-type is a permutation is at most $m! \cdot 2^m$, it follows that the total number of colors is at most $m! \cdot 2^m \cdot \lceil \frac{10m}{\gamma} + 1 \rceil^{(m+1)} \leq (\frac{100m}{\gamma})^{2m}$.

**Ramsey.** We next apply Ramsey Theorem to derive a large $\mathcal{X}' \subseteq \mathcal{X}_n$ such that every subset $D \subseteq \mathcal{X}_n$ of size $m+1$ has the same color. Later we will argue that $A$ is $\gamma$-approximately homogeneous with respect to $\mathcal{X}'$ which will finish the proof.

We will use the following quantitative version of Ramsey Theorem due to [14] (see also the book [15], or Theorem 10.1 in the survey by [26]). Here, the *tower function* $\mathbf{twr}_k(x)$ is defined by the recursion

$$\mathbf{twr}^{(i)}x = \begin{cases} x & i = 1, \\ 2^{\mathbf{twr}(i-1)(x)} & i > 1. \end{cases}$$

**Theorem 3** (Ramsey Theorem [14]). *Let $s > t \geq 2$ and $q$ be integers, and let*

$$N \geq \mathbf{twr}_t(3sq \log q).$$

*Then, for every coloring of the subsets of size $t$ of a universe of size $N$ using $q$ colors there is a homogeneous subset[9] of size $s$.*

Stated differently, Theorem 3 guarantees the existence of a homogeneous subset of size

$$\frac{\log^{(t-1)}(N)}{3q \log q}. \tag{14}$$

Thus, by plugging $q := (\frac{10m}{\gamma})^{2m}, t := m + 1, N := n$ in Equation (14) we get a homogeneous set $\mathcal{X}' \subseteq \mathcal{X}_n$ of size

$$|\mathcal{X}'| \geq \frac{\log^{(m)}(n)}{3(\frac{10m}{\gamma})^{2m} \cdot 2m \log(\frac{10m}{\gamma})} \geq \frac{\log^{(m)}(n)}{(\frac{10m}{\gamma})^{3m}}.$$

**Wrapping-up.** It remains to show that $A$ is $\gamma$-approximately homogeneous with respect to $\mathcal{X}'$. By the construction of $\mathcal{X}'$ there exist a specific color

$$L = \left\{ (p_i^{\pi,\bar{y}})_{i=0}^m : (\pi, \bar{y}) \text{ is an equivalence-type s.t. } \pi \text{ is a permutation} \right\}$$

such that $c(D) = L$ for every $D = \{x_0 < \ldots < x_m\} \subseteq \mathcal{X}'$. We need to show that for every pair of equivalent samples $S', S''$ whose order-type is a permutation and for every $x \in \mathcal{X}' \setminus \underline{S}, x' \in \mathcal{X}' \setminus \underline{S'}$ such that $\mathsf{pos}(x, S) = \mathsf{pos}(x', S')$:

$$\left| \Pr_{h \sim Q_S}[h(x) = 1] - \Pr_{h' \sim Q_{S'}}[h'(x') = 1] \right| \leq \frac{\gamma}{5m}.$$

Let $(\pi, \bar{y})$ be an equivalence-type such that $\pi$ is a permutation, let $S$ be any sample whose equivalence-type is $(\pi, \bar{y})$, and let $x \in \mathcal{X}' \setminus \bar{S}$. Consider the set $D = \{x_j : j \leq m\} \cup \{x\}$ and set $i = \mathsf{pos}(x, S)$. By the definition of $D_{-i}^{\pi,\bar{y}}$, we have $D_{-i}^{\pi,\bar{y}} = S$ and hence by the definition of $p_i^{\pi,\bar{y}}$ we have

$$\left| \Pr_{h \sim Q_S}[h(x) = 1] - p_i^{\pi,\bar{y}} \right| = \left| \Pr_{h \sim Q_{-i}^{\pi,\bar{y}}}[h(x) = 1] - p_i^{\pi,\bar{y}} \right| \leq \frac{\gamma}{10m}.$$

Since the latter holds for every sample $S$ whose order type is $(\pi, \bar{y})$ and every $x \notin \bar{S}$, it follows that for every pair of samples $S, S'$ whose order-type is $(\pi, \bar{y})$ and every $x \in \mathcal{X}' \setminus \underline{S}, x' \in \mathcal{X}' \setminus \underline{S}'$ such that $\mathsf{pos}(x, S) = \mathsf{pos}(x', S')$:

$$\left| \Pr_{h \sim Q_S}[h(x) = 1] - \Pr_{h' \sim Q_{S'}}[h'(x') = 1] \right| \leq$$

$$\left| \Pr_{h \sim Q_S}[h(x) = 1] - p_i^{\pi, \bar{y}} \right| + \left| \Pr_{h \sim Q_S}[h(x) = 1] - p_i^{\pi, \bar{y}} \right| \leq \frac{\gamma}{10m} + \frac{\gamma}{10m} = \frac{\gamma}{5m},$$

where $i := \mathsf{pos}(x, S) = \mathsf{pos}(x', S')$. This finishes the proof. $\square$

### A.3 Proof of Lemma 2

**Notation.** We will assume without loss of generality that $I = \{1, 2, 3, ..., |I|\}$. Also, to simplify the presentation, we will assume that $|I|$ is a power of 2, i.e. $|I| = 2^b$ for some $b \in \mathbb{N}$. (Removing this assumption is straight-forward, but complicates some of the notation.)

**Overview.** Let $P$ be an arbitrary prior supported on $\{\pm 1\}^I$. Our goal is to show that at least $|I|/4$ of all $\hat{x}$'s in $I$ satisfy

$$\mathrm{KL}\left(Q_{\hat{x}} \| P\right) \geq \Omega\left((q_2 - q_1)^2 \frac{\log|I|}{\log\log|I|}\right) = \Omega\left((q_2 - q_1)^2 \frac{b}{\log(b)}\right).$$

The proof strategy is to bound from below $\mathrm{KL}\left(Q_{\hat{x}}^m \| P^m\right)$, where $m$ is sufficiently small; the desired lower bound then follows from the chain rule:

$$\mathrm{KL}\left(Q_{\hat{x}} \| P\right) = \frac{1}{m}\mathrm{KL}\left(Q_{\hat{x}}^m \| P^m\right).$$

Obtaining the lower bound with respect to the $m$-fold products is the crux of the proof. In a nutshell, we will exhibit events $E_{\hat{x}}$ such that for every $\hat{x} \in I$, $Q_{\hat{x}}^m(E_{\hat{x}}) \geq 1/2$, , but for $|I|/4$ of the $\hat{x}$'s, $P^m(E_{\hat{x}})$ is tiny. This implies a lower bound on $\mathrm{KL}\left(Q_{\hat{x}}^m \| P^m\right)$ since

$$\mathrm{KL}\left(Q_{\hat{x}}^m \| P^m\right) \geq \mathrm{KL}\left(Q_{\hat{x}}^m(E_{\hat{x}}) \| P^m(E_{\hat{x}})\right),$$

by the data-processing inequality.

**Construction of The Events $E_{\hat{x}}$.** For every Gibbs-classifier $Q \in \{Q_{\hat{x}} : \hat{x} \in I\} \cup \{P\}$ define its *rounded-hypothesis* $\mathbf{h}_Q : X \to \{\pm 1\}$ as follows:

$$\mathbf{h}_Q(x) = \begin{cases} -1 & \mathbb{E}_{h \sim Q_{\hat{x}}}[h(x)] \leq \frac{q_1 + q_2}{2}, \\ +1 & \mathbb{E}_{h \sim Q_{\hat{x}}}[h(x)] > \frac{q_1 + q_2}{2}. \end{cases}$$

To simplify notation, let $\mathbf{h}_{\hat{x}} = \mathbf{h}_{Q_{\hat{x}}}$. Note that by the assumption of Lemma 2:

$$\mathbf{h}_{\hat{x}}(x) = \begin{cases} -1 & x < \hat{x}, \\ +1 & x > \hat{x}. \end{cases} \tag{15}$$

In words, each $\mathbf{h}_{\hat{x}}$ is a threshold with a sign-change either right before $\hat{x}$ or right after it. Next, given $\mathbf{h} : I \to \{\pm 1\}$, consider the following iterative process which applies binary-search on $\mathbf{h}$ towards detecting a pair of subsequent coordinates which contain a sign-change.

---

**Binary-Search**

Input: $\mathbf{h} : I \to \{\pm 1\}$.

    1. Set $I_0 = [a_0, b_0]$, where $a_0 = 0, b_0 = |I| = 2^b$.

    2. For $j = 0, \ldots$

        (a) If $|I_j| \leq 2$ then output $I_j$.

        (b) Query the coordinate $\mathbf{h}(m_j)$, where $m_j = \frac{a_j + b_j}{2}$.

        (c) If $\mathbf{h}(m_j) = +1$ then set $a_{j+1} = a_j, b_{j+1} = m_j$,

        (d) Else, set $a_{j+1} = m_j + 1, b_{j+1} = b_j$.

---

The following observations follow from the standard analysis of binary-search.

1. The process ends after $b - 1$ iterations and each of the points $m_j$ queried in Item (b) are even numbers.

2. If the process is applied on a threshold $\mathbf{h}$ which changes sign from $-$ to $+$ between $x$ and $x + 1$ then the output interval $I_{out}$ is $\{x, x + 1\}$. Thus, by Equation (15), if we apply this process on $\mathbf{h} = \mathbf{h}_{\hat{x}}$ then $\hat{x} \in I_{out}$.

Given a sequence of hypotheses $h_1, \ldots, h_m : I \to \{\pm 1\}$, define the *empirical rounded-hypothesis* $\mathbf{h}_{h_{1:m}}$ by:

$$\mathbf{h}_{h_{1:m}}(x) = \begin{cases} -1 & \frac{1}{m} \sum_{i=1}^{m} \mathbf{1}[h_i(x) = 1] \le \frac{q_1 + q_2}{2}, \\ +1 & \frac{1}{m} \sum_{i=1}^{m} \mathbf{1}[h_i(x) = 1] > \frac{q_1 + q_2}{2}. \end{cases}$$

Consider $h_1, \ldots, h_m \sim Q_{\hat{x}}$ for an odd $\hat{x} \in I$. The following claim shows that with high probability, applying the binary search on $\mathbf{h}_{h_{1:m}}$ yields an output interval $I_{out}$ such that $\hat{x} \in I_{out}$.

**Claim 4.** *Let $\hat{x} \le 2^b$ be an odd number. Let $J_{out}$ denote the interval outputted by applying the binary search on $\mathbf{h}_{\hat{x}}$ and let $I_{out}$ denote the interval outputted by applying the binary search on $\mathbf{h}_{h_{1:m}}$, where $h_1, \ldots h_m \sim Q_{\hat{x}}$ are drawn independently. Then,*

$$\Pr_{h_1 \ldots h_m \sim Q_{\hat{x}}^m}[I_{out} \ne J_{out}] \le b \cdot \exp\left(-\frac{m}{2}(q_2 - q_1)^2\right).$$

*In particular, if $m = \frac{2(\ln(b) + 2)}{(q_2 - q_1)^2}$ then $\Pr[\hat{x} \notin I_{out}] \le \frac{1}{2}$.*

*Proof.* Let $x_1, \ldots x_2, \ldots, x_{b-1}$ be the coordinates queried by the binary search on $J_{out}$. We will show that with high probability $\mathbf{h}_{h_{1:m}}(x_i) = \mathbf{h}_{\hat{x}}(x_i)$ for every $i$, which implies that $J_{out} = I_{out}$. Let $i \le b - 1$ and define

$$\mu_i = \mathbb{E}_{h \sim Q_{\hat{x}}}[\mathbf{1}[h(x_i) = +1]] = \Pr_{h \sim Q_{\hat{x}}}[h(x_i) = +1].$$

Note that $\hat{x} \ne x_i$ (because $x_i$ is even and $\hat{x}$ is odd). Therefore, by the assumption of Lemma 2:

$$\mu_i \begin{cases} \le \frac{q_2 + q_1}{2} - \frac{q_2 - q_1}{4} & x_i < \hat{x}, \\ \ge \frac{q_2 + q_1}{2} + \frac{q_2 - q_1}{4} & x_i > \hat{x}. \end{cases}$$

Hence, by a Chernoff bound:

$$\Pr_{h_1 \ldots h_m}[\mathbf{h}_{h_{1:m}}(x_i) \ne \mathbf{h}_{\hat{x}}(x_i)] \le \Pr_{h_1 \ldots h_m}\left[\frac{1}{m} \sum_{j=1}^{m} \mathbf{1}[h_j(x_i) = 1] \ge \mu_i + \frac{q_2 - q_1}{4}\right]$$

$$\le \exp\left(-\frac{m}{2}(q_2 - q_1)^2\right) \qquad \text{(Chernoff Bound)}$$

Thus, by taking a union bound over all $i \le b - 1$ it follows that $\mathbf{h}_{h_{1:m}}(x) = \mathbf{h}_{\hat{x}}(x)$ for every $i \le b - 1$ with probability at least $1 - \log(|I|) \cdot \exp(-\frac{m}{2}(q_2 - q_1)^2)$. In particular, with the above probability we have that $J_{out} = I_{out}$.

Lastly, assume $m = \frac{2(\ln(b) + 2)}{(q_2 - q_1)^2}$. Then, $b \cdot \exp(-\frac{m}{2}(q_2 - q_1)^2) \le 1/2$, and therefore $\Pr[J_{out} = I_{out}] \ge \frac{1}{2}$. Since $\mathbf{h}_{\hat{x}}$ is a threshold which changes sign either right before $\hat{x}$ or right after $\hat{x}$, it follows that $\hat{x} \in J_{out}$, and therefore $\Pr[\hat{x} \in I_{out}] \ge 1/2$. $\square$

We are now ready to define the events $E_{\hat{x}}$. Set $m = \frac{2(\ln(b) + 2)}{(q_2 - q_1)^2}$, according to Claim 4, and let $E_{\hat{x}}$ denote the event that $\hat{x} \in I_{out}$. That is, $E_{\hat{x}}$ is the set of all sequences $h_1, \ldots h_m$ such that $\hat{x} \in I_{out}$, where $I_{out}$ is the interval outputted by the binary-search on $\mathbf{h}_{h_{1:m}}$. Thus, Claim 4 says that $Q_{\hat{x}}^m(E_{\hat{x}}) \ge \frac{2}{3}$ for an odd $\hat{x}$.

**Bounding the KL-divergence.** We next use the events $E_{\hat{x}}$ to lower bound $\mathrm{KL}\left(Q_{\hat{x}}\|P\right)$:

$$\mathrm{KL}\left(Q_{\hat{x}}\|P\right) = \frac{1}{m}\mathrm{KL}\left(Q_{\hat{x}}^m\|P^m\right) \qquad \text{(Chain Rule)}$$

$$\geq \frac{1}{m}\mathrm{KL}\left(Q_{\hat{x}}^m(E_{\hat{x}})\|P^m(E_{\hat{x}})\right) \qquad \text{(Data Processing Ineq.)}$$

$$\geq \frac{1}{m}\left(-\frac{2}{3}\log\left(\frac{2}{3}\right) - \frac{1}{3}\log\left(\frac{1}{3}\right) - \frac{2}{3}\log\left(P^m(E_{\hat{x}})\right)\right)$$

$$\geq \frac{-\log\left(P^m(E_{\hat{x}})\right) - 1}{2m}$$

Therefore, to lower bound $\mathrm{KL}\left(Q_{\hat{x}}\|P\right)$ it suffices to shows that $P^m(E_{\hat{x}})$ is small. We next establish this for $1/4$ of the $\hat{x}$'s in $I$. Note that whenever $\hat{x}_1, \hat{x}_2 \in I$ are odd and distinct then $E_{\hat{x}_1} \cap E_{\hat{x}_2} = \emptyset$. Indeed, this follows since the outputted interval $I_{out}$ is of size $\leq 2$ and hence contains at most one odd number. Thus,

$$\sum_{\hat{x} \text{ is odd}} P^m(E_{\hat{x}}) \leq 1.$$

In particular, since there are $2^{b-1}$ odd numbers in $I$, at least $1/2$ of them must satisfy $P^m(E_{\hat{x}}) \leq \frac{1}{2^{b-2}}$. Taken together we obtain that at least $1/4$ of all $\hat{x} \in I$ satisfy:

$$\mathrm{KL}\left(Q_{\hat{x}}\|P\right) \geq \frac{b-2-1}{2m}$$

$$= \frac{b-1}{2\frac{2(\ln(b)+2)}{(q_2-q_1)^2}} = \Omega\left((q_2-q_1)^2\frac{b}{\log(b)}\right),$$

which finishes the proof of Lemma 2

$\square$