[Reviews · NeurIPS 2020]

Review 1

Summary and Contributions: The paper demonstrates a scenario -- namely learning initial segments -- where any PAC-Bayes bound is vacuous, although the standard VC bound works just fine. In particular, the lower bound depends (although mildly) on the size of the domain. The lower bound is proved for any algorithm, not just usual Gibbs classifiers (e.g., exponential or Gaussian). The proof deviates from the usual minimax arguments where a hard distribution is found; here a more involved argument is needed.

Strengths: The contributions of the paper can be conceptually divided into two parts: (i) proving the lower bound for a specific but natural algorithm and (ii) generalizing the argument to all algorithms. Even the first part is quite illuminating: it shows that the KL divergence of the prior and posterior is Omega(log|n|) where n is the domain size. This shows a fundamental limitation of PAC-Bayes approach (which is being used frequently these days) The second part generalizes this to arbitrary algorithms, with the cost of having a weaker lower bound Omega(log*(n)). The technique of using Ramsey theory in the argument is quite interesting (although it is based on a similar argument in [1]) and may have other applications. As far as I know this is the first paper that shows the limitations of the PAC Bayes framework for classification. Moreover, the techniques used for the proof are novel, sophisticated and potentially useful in other scenarios.

Weaknesses: In practice, people sometimes (often?) discritize the parameter space and then use the PAC bound on that discritized hypothesis class. This improves the bound significantly in practice (although in theory the discritization can hurt in the worst case), and the lower bound may not be that powerful for the finite hypothesis class that is built using discritization. More discussion on this can help to increase the applicability of this lower bound to practical situations.

Correctness: The claims and arguments seem sound. I suggest that the authors elaborate more on data dependent priors and why it does not affect the arguments.

Clarity: The paper is very well written. Few typos: + page 2, "an hypothesis" + page 5, "there exist an index i \leq m\prime" (prime should be removed?) + lemma 2, KL is missing in the main statement, "with probability at least 1/4:..."

Relation to Prior Work: The relation to prior work is very well elaborated.

Reproducibility: Yes

Additional Feedback:


Review 2

Summary and Contributions: 1. The paper demonstrates (via lower bounds on the standard PAC-Bayes bounds) that uniform learning cannot be explained by the standard PAC-Bayes bounds 2. The authors introduce a new (to PAC-Bayes) proof technique to demonstrate this using Ramsey Theory. The connection to Ramsey Theory is quite interesting as an avenue for future work on lower bounds.

Strengths: 1. The claims appear to be sound. 2. The proof technique is interesting and could likely be modified and used for more problems in the future for deriving lower bounds. This will give the paper some long term impact. 3. While the overall result was not surprising at first, that the choice of adversarial distribution does not depend on the prior IS very surprising and interesting. The authors should aim to highlight this more and explain why this is more difficult than allowing the adversarial distribution to depend on P, because it dramatically effected how excited I was about the result.

Weaknesses: 1. The result is not surprising at first glance, since PAC-Bayes results are essentially non-uniform learning bounds (the sample complexity depends on the identity of the best hypothesis) rather than uniform learning bounds (sample complexity independent of the identity of best hypothesis), where the hypothesis space can be seen as "gradated" based on the prior probability intensity. 2. Should explain earlier and more intuitively why having the adversarial distribution choice not depend on P is challenging and more interesting.

Correctness: The claims appear to be correct.

Clarity: 1. The paper is very well written and quite easy to read. I think it would even be easily understood by readers with less background knowledge on learning theory and PAC-Bayes.

Relation to Prior Work: 1. The relation to prior work on PAC-Bayes is well motivated. 2. I am not very familiar with Ramsey theory so I cannot assess whether the corresponding relationship with literature is discussed adequately.

Reproducibility: Yes

Additional Feedback: 1. Is it possible to do something similar to show an example where PAC-Bayes does cannot even lead to a non-uniform learning bound? That would be a more surprising result! 2. The relationship between uniform learning, PAC-Bayes, non-uniform learning, and these new lower bounds should be further explored in the set up of the problem. This can then highlight how the different order of quantifiers or sups/infs lead to different strengths of the conclusions, and how PAC-Bayes is more like non-uniform learning to begin with. This could easily be used to argue that an adversarial distribution can be constructed with knowledge of the prior. Then the novelty of this work can be framed in terms of constructing adversarial distributions WITHOUT knowledge of the prior (changes the order of quantifiers, or the order of sup/inf/sup). 3. The score I gave (7) is based on the current state of the manuscript. If the authors adequately addressed the feedback I gave in the rebuttal period, I would likely raise my score to an 8. To go above an 8, I would need to be shown that there is a significant contribution that I have failed to acknowledge or that I've misunderstood something which materially affects the impact that the work will have. POST REBUTTAL COMMENTS: The authors essentially agreed with the critical points I've made, though they have not given concrete evidence of changes they will make. I increased my score from 7 to 8 based on the good-faith promise of an adequate discussion of the relationship to non-uniform learning. Whether their findings in this regard end up being positive or negative, I think it is an important facet to the interpretation of this work,


Review 3

Summary and Contributions: The central theoretical result of this paper is an impossibility theorem. It shows that McAllester's PAC-Bayes bound cannot explain the small sample complexity of a one-dimensional linear classification learning problem.

Strengths: Rigorously stating the strengths and the limitations of learning theories is crucial for ensuring that our science evolves on solid bases. It seems that the paper contributes to the PAC-Bayes framework in this direction.

Weaknesses: The paper is technically heavy for my expertise, so I can only raise questions about its content. Might they be naive, discussing them in the paper would help other readers to understand the scope of this work. A first concern is about the fact that the paper presents solely (Theorem 1) the PAC-Bayes bound of McAllester (1999), converging at rate sqrt(1/m). Since this pioneer work, many variations on the PAC-Bayes bounds have been proposed. Notably, Seeger (2002)'s and Catoni (2007)'s bounds are known to converge at rate 1/m when the empirical risk is zero (see also Guedj (2019) for a up-to-date overview of PAC-Bayes literature). Is the given impossibility theorem and its proof remain the same for these settings? I also wonder if the issue with the PAC-Bayes theorem comes from the fact that, by considering the Gibbs predictor (an average of predictors), the aim of the PAC-Bayes estimator is to model a [0,1] target: P(y=1|x), as stated by the paragraph of Lines 238-242. Would it be natural (or not) that such a regressor is not as efficient to model the {0,1} thresholded distribution than if the predictor hypothesis class is restricted to linear binary classifiers? Related to the latter point, of potential interest is the line of works adapting the PAC-Bayes results to "weighted majority vote" (e.g., Germain et al. 2015), where one wants to bound the loss of the so-called Bayes classifier B_Q(x) = sgn(E_Q(h(x)) \in {-1,1} instead of the Gibbs randomized classifier with E G_Q(x) = E_Q(h(x)) \in [-1,1]. Note that the latter expected output of the Gibbs classifier, E G_Q(x), can be interpreted as a regressor, References: Catoni (2007). PAC-Bayesian Supervised Classification: The Thermodynamics of Statistical Learning. Institute of Mathematical Statistics, Germain, Lacasse, Laviolette, Marchand, Roy (2015). Risk bounds for the majority vote: from a PAC-Bayesian analysis to a learning algorithm. JMLR Guedj 2019, A Primer on PAC-Bayesian Learning. Proceedings of the second congress of the French Mathematical Society. Seeger (2002). PAC-Bayesian generalization bounds for Gaussian processes.JMLR

Correctness: Unfortunately, I don't have the expertise to validate the proof in a reasonable time. I could only perceive that the work seems to have been done rigorously,

Clarity: The paper is well-written. I first sight, I was annoyed by the way the authors refer to the proof in supplementary material, i.e. "Proofs are provided in the full version". This suggests that the submitted manuscript is a condensed version of another paper, while it must be a standalone piece of work. Finally this is the case, as the "full version" looks alike the manuscript, augmented by a "Proof" section. In definitive, this is not a major issue.

Relation to Prior Work: See the references cited in the above "Weaknesses" section.

Reproducibility: Yes

Additional Feedback: Several published references are cited as arXiv: [2] is published in ICML 2018; [11] is published in UAI 2017; [22] is published in ICLR 2018. Please cite the peer-reviewed versions. === Post rebuttal === I am disappointed that the rebuttal did not discuss most of my comments. In particular, the one about the fact that PAC-Bayes "Gibbs predictor" is aiming for a [0,1] target instead of a {0,1} classification. I was wondering if it could explain the stated limitation, while one could also argue that it is an asset of the theory. Note that I do not consider that it would diminish the value of the work, but it would be interesting to be discussed. Maybe I missed something and/or my comment was unclear, but given that the authors choose to ignore it, I am not inclined to change my score. That being said, the paper is interesting, well written, and would generate discussion. I still have difficulty to validate the proof details, but I trust R1 and R2 on this aspect.


Review 4

Summary and Contributions: This paper presents a limitation for the PAC-Bayes framework, that is, there are classes that are learnable but this cannot be proved using a PAC-Bayes analysis. Since these exists a realizable distribution for which the PAC-Bayes bound is arbitrarily large.

Strengths: This paper firstly gives the main results and then takes much effort to prove its findings in a detailed and ordered manner. It is the first to investigate the limitation of PAC-Bayes framework which is used in the class of 1-dimensional thresholds. Therefore, the paper provides a limited significant and novel contribution and is relevant to the NeurIPS.

Weaknesses: This paper provides the evidence that the PAC-Bayes framework is useless in proving the learnability of linear classification in 1D. The method proposed in this paper is limited in the simple linear classification and can be seen as a specific counter-example of PAC-Bayes analysis. Hence, it is suggested to explore whether there exist other learnable tasks which cannot be proved with a PAC-Bayes analysis and specify the scope of effectiveness of PAC-Bayes analysis.

Correctness: Although the main body provides the proof-sketch, there still are confusing parts. It is better to explain what the “hard” distribution is in detail and illustrate the connection between defining the Homogeneity and detecting the hard distribution. The ending of the proof is ambiguous to derive the final conclusion, so it is suggested that providing the complete proof in the main body rather than in the supplementary.

Clarity: The paper is well written but there exist some mistakes. First of all, the last line of Theorem 1 is not contained the environment of Theorem. Secondly, there are some grammatical mistakes which are listed as follow: 1) “label” in line 15 should be “labels”. 2) “follow” in line 44 should be “follows”. 3) “an hypothesis” in line 51, 64 and 75 should be “a hypothesis”. 4) “ m’ ” in line 235 should be “m”.

Relation to Prior Work: Yes, the paper provides detailed discussions of previous contributions.

Reproducibility: Yes

Additional Feedback: The paper revealed that "For any algorithm that learns 1-dimensional linear classifiers there exists a (realizable) distribution for which the PAC-Bayes bound is arbitrarily large". I still wonder that, could this be seen from real data sets? Othervise, it is purely theoretical and perhaps can not guide the practical applications of the theory since some ditributions rarely appear in reality. Could you report some real experiments on the theory? I really hope to see such results.

[Author Response · NeurIPS 2020]

We thank the reviewers for taking the time to read this work and for their thoughtful reviews, Below
we address some of the specific comments raised by the reviewers..

## Reviewer 1

• **The ideas presented in the paper rely heavily on the choice of hypothesis class** Note that the
result immediately applies to any class that contains 1d thresholds as a subclass. In particular it
extends to linear classifiers in any dimension (which is often used in practice).

• **Also, in practice, people sometimes (often?) discritize the parameter space and then use the
PAC bound on that discritized hypothesis class.** Here, our result implies that the PAC Bayes
guarantee will deteriorate with the precision of the discretization (as opposed to VC bounds).

## Reviewer 2

• **Should explain earlier and more intuitively why having the adversarial distribution choice
not depend on P is challenging and more interesting** We did try to address this in the paper, but
evidently it is not emphasized enough. We will gladly highlight it earlier and in more detail.
We address this fact by stating that the prior may depend on the distribution (which is the same
as saying that the bounds hold for a distribution which is chosen independent of the prior). E.g.
in the introduction starting line 34 "We stress that...", line 40: "We emphasize..." Line 84:
"Remarkably ....". Then, in terms of the technical challenge in lines 121 to lines 127 we explain
why choosing the prior as a function of the distribution leads to the main technical difficulty.
In any case we will definitely highlight this in the full version and rearrange things.

• **Is it possible to show non-uniform learning bound?** This is an excellent question. We will give
this some thought and see if our techniques imply such bounds. In any case we will discuss this
question in the final version. Thanks!

• **The relationship between uniform learning, PAC-Bayes, non-uniform learning, and these
new lower bounds should be further explored in the set up of the problem.** We will follow
this advice and highlight the points raised.

## Reviewer 3

• **Notably, Seeger (2002)'s ... Is the given impossibility theorem and its proof remain the same
for these settings?** We note that the adversarial distribution we choose is realizable (as stated,
see corollary 1 and thm2). So the result holds for this setting as well. We will gladly add further
discussion on the implication of our result to these more modern bounds. Thanks for suggesting
this.

• **Several published references are cited as arXiv** Thanks, we will correct this.

## Reviewer 4

• **The method proposed in this paper is limited in the simple linear classification and can be
seen as a specific counter-example of PAC-Bayes analysis.** The contribution of this paper is an
impossibility (hardness) result for the PAC Bayes framework. By definition, hardness results follow
by exhibiting a specific counter-example, and we don't understand in what sense it is a weakness.

• **It is suggested to explore whether there exist other learnable tasks which cannot be proved
with a PAC-Bayes analysis and specify the scope of effectiveness of PAC-Bayes analysis.**
A natural future direction is indeed to characterize which classes exactly are amenable to PAC-Bayes
analysis (see also our answer to reviewer 1 for a related question). We also discuss this future
challenge in section 5, where we highlight recent results that might hint that understanding the
role of Littlestone dimension can help to characterize the classes that are amenable to PAC-Bayes
analysis. But again, we are not sure what is the weakness here.

• **so it is suggested that providing the complete proof in the main body rather than in the
supplementary.**
With the 8-page limitation of Neurips this is not reasonable.

• Overall we could not understand what is the justification for the low score provided by the reviewer.
We hope the reviewer will reconsider her/his score.



[Meta-Review · NeurIPS 2020]

This paper points out a fundamental limitation of the PAC-Bayes framework; namely, that there exist problems/distributions that are known to be learnable for which PAC-Bayes bounds fail to give meaningful generalization guarantees. This is an important (albeit negative) result that furthers our understanding of this powerful theoretical framework. The reviews agree that the subject is timely and compelling; the paper is well written and its results are more or less presented with clarity. The reviews also raise some valid concerns: 1) The chosen PAC-Bayes bound is by far not the tightest, and that tighter, more recent bounds should be considered. The authors responded by saying that even Seeger's bound can fail, but this should be discussed in detail in the paper. 2) The setting considered -- a 1-D classification problem -- is not very interesting from a practical perspective. That said, most reviewers also felt that this is a minor concern, and the authors point out that, technically, the results hold for any class that contains 1-D classifiers as a subclass (such as N-D classifiers). Most reviewers remained positive on the paper after the author response, but they were also disappointed that the response did not address some of their concerns. That said, with a 1 page limit, it is impossible to address everything. I strongly encourage the authors to incorporate *all* of the reviewers' feedback into the final version of the paper.